# Unsupervised Image Classification for Deep Representation Learning

Paper ID Anonymous

**Abstract.** Deep clustering against self-supervised learning (SSL) is a very important and promising direction for unsupervised visual representation learning since it requires little domain knowledge to design pretext tasks. However, the key component, embedding clustering, limits its extension to the extremely large-scale dataset due to its prerequisite to save the global latent embedding of the entire dataset. In this work, we aim to make this framework more simple and elegant without performance decline. We propose an unsupervised image classification framework without using embedding clustering, which is very similar to standard supervised training manner. For detailed interpretation, we further analyze its relation with deep clustering and contrastive learning. Extensive experiments on ImageNet dataset have been conducted to prove the effectiveness of our method. Furthermore, the experiments on transfer learning benchmarks have verified its generalization to other downstream tasks, including multi-label image classification, object detection, semantic segmentation and few-shot image classification.

**Keywords:** Unsupervised Learning, Representation Learning

## 1  Introduction

Deep convolutional neural networks (CNN) [16, 19, 5] had been applied to many computer vision applications [14, 26, 25] due to their powerful representational capacity. The normal working flow is to pretrain the networks on a very large-scale dataset with annotations like ImageNet [31] and then transfer to a small dataset via fine-tuning. However, the dataset collection with manually labelling for pre-training is strongly resource-consuming, which draws lots of researchers' attention to develop unsupervised representation learning approaches.

Among the existing unsupervised learning methods, self-supervision is highly sound since it can directly generate supervisory signal from the input images, like image inpainting [8, 30] and jigsaw puzzle solving [28]. However, it requires rich empirical domain knowledge to design pretext tasks and is not well-transferred to downsteam tasks. Compared with this kind of self-supervised approaches, DeepCluster is a simple yet effective method which involves litter domain knowledge. It simply adopts embedding clustering to generate pseudo labels by capturing the manifold and mining the relation of all data points in

**Fig. 1.** The pipeline of unsupervised image classification learning. The black and red arrows separately denote the processes of pseudo-label generation and representation learning. These two processes are alternated iteratively. For efficient implementation, the psuedo labels in current epoch are updated by the forward results from the previous epoch which means our training framework is twice faster than DeepCluster.

the dataset. This process is iteratively alternated with an end-to-end representation learning which is exactly the same with supervised one. However, along with the advantage brought by embedding clustering, an obvious defect naturally appears that the latent embedding of each data point in the dataset should be saved before clustering, which leads to extra memory consumption linearly growing with the dataset size. It makes it difficult to scale to the very large-scale datasets. Actually, this problem also happens in the work of DeeperCluster [3], which uses distributed $k$-means to ease the problem. However, it still did not solve the problem in essence. Also, the data points in most of datasets are usually independently identically distributed ($i.i.d$). Therefore, building a framework analogous to DeepCluster, we wonder if we can directly generate pseudo class ID for each image without explicitly seeing other images and take it as an image classification task for representation learning.

The answer is excitedly YES! We integrate both the processes of pseudo label generation and representation learning into an unified framework of image classification. Briefly speaking, during the pseudo label generation, we directly feed each input image into the classification model with softmax output and pick the class ID with highest softmax score as pseudo label. It is very similar to the inference phase in supervised image classification. After pseudo class IDs are generated, the representation learning period is exactly the same with supervised training manner. These two periods are iteratively alternated until convergence. A strong concern is that if such unsupervised training method will be easily trapped into a local optima and if it can be well-generalized to other downstream tasks. In supervised training, this problem is usually solved by data augmentation which can also be applied to our proposed framework. It is worth noting that we not only adopt data augmentation in representation learning but also in pseudo label generation. It can bring disturbance to label assignment and make the task more challenging to learn data augmentation agnostic features. The entire pipeline is shown in Fig.1. To the best of our knowledge, this unsupervised framework is the closest to the supervised one compared with other existing works. Since it is very similar to supervised image classification, we name our method as *Unsupervised Image Classification* (UIC) correspondingly. For sim-

plicity, without any specific instruction, *clustering* in this paper only refers to embedding clustering via $k$-mean, and *classification* refers to CNN-based classification model with cross-entropy loss function.

To further explain why UIC works, we analyze its hidden relation with both deep clustering and contrastive learning. We point out that UIC can be considered as a special variant of them. We hope our work can bring a deeper understanding of deep clustering series work to the self-supervision community.

We empirically validate the effectiveness of UIC by extensive experiments on ImageNet. The visualization of classification results shows that UIC can act as clustering although lacking explicit clustering. We also validate its generalization ability by the experiments on transfer learning benchmarks. All these experiments indicate that UIC can work comparable with deep clustering. To summarize, our main contributions are listed as follows:

- A simple yet effective unsupervised image classification framework is proposed for visual representation learning, which can be taken as a strong prototype to develop more advanced unsupervised learning methods.
- Our framework simplifies DeepCluster by discarding embedding clustering while keeping no performance degradation and surpassing most of other self-supervised learning methods. We demonstrate that embedding clustering is not the main reason why DeepCluster works.
- Our training framework is twice faster than DeepCluster since we do not need an extra forward pass to generate pseudo labels.
- We connect our proposed unsupervised image classification with deep clustering and contrastive learning for further interpretation.

## 2   Related Work

### 2.1   Self-supervised learning

Self-supervised learning is a major form of unsupervised learning, which defines pretext tasks to train the neural networks without human-annotation, including image inpainting [8, 30], automatic colorization [23, 39], rotation prediction [13], cross-channel prediction [40], image patch order prediction [28], and so on. These pretext tasks are designed by directly generating supervisory signals from the raw images without manually labeling, and aim to learn well-pretrained representations for downstream tasks, like image classification, object detection, and semantic segmentation. Recently, contrastive learning [33, 15, 18, 29] is developed to improve the performance of self-supervised learning. Its corresponding pretext task is that the features encoded from multi-views of the same image are similar to each others. The core insight behind these methods is to learn multi-views invariant representations. This is also the essence of our proposed method.

### 2.2   Clustering-based methods

Clustering-based methods are mostly related to our proposed method. Coates et al. [7] is the first to pretrain CNNs via clustering in a layer-by-layer manner.

The following works [37, 36, 24, 2] are also motivated to jointly cluster images and learn visual features. Among them, DeepCluster [2] is one of the most representative methods in recent years, which applies $k$-means clustering to the encoded features of all data points and generates pseudo labels to drive an end-to-end training of the target neural networks. The embedding clustering and representation learning are iterated by turns and contributed to each other along with training. Compared with other SSL methods with fixed pseudo labels, this kind of works not only learn good features but also learn meaningful pseudo labels. However, as a prerequisite for embedding clustering, it has to save the latent features of each sample in the entire dataset to depict the global data relation, which leads to excessive memory consumption and constrains its extension to the very large-scale datasets. Although another work DeeperCluster [3] proposes distributed $k$-means to ease this problem, it is still not efficient and elegant enough. Another work SelfLabel [1] treats clustering as a complicated optimal transport problem. It proposes label optimization as a regularized term to the entire dataset to simulate clustering with the hypothesis that the generated pseudo labels should partition the dataset equally. However, it is hypothesized and not an $i.i.d$ solution. Interestingly, we find that our method can naturally divide the dataset into nearly equal partitions without using label optimization.

## 3 Methods

### 3.1 Preliminary: Deep Clustering

Before introducing our proposed unsupervised image classification method, we first review deep clustering to illustrate the process of pseudo label generation and representation learning, from which we analyze the disadvantages of embedding clustering and dig out more room for further improvement.

**Pseudo Label Generation.** Most self-supervised learning approaches focus on how to generate pseudo labels to drive unsupervised training. In deep clustering, this is achieved via $k$-means clustering on the embedding of all provided training images $X = x_1, x_2, ..., x_N$. In this way, the images with similar embedding representations can be assigned to the same label.

Commonly, the clustering problem can be defined as to optimize cluster centroids and cluster assignments for all samples, which can be formulated as:

$$\min_{C \in \mathbb{R}^{d \times k}} \frac{1}{N} \sum_{n=1}^{N} \min_{y_n \in \{0,1\}^k \ s.t. y_n^T \mathbf{1}_k = 1} \| C_{y_n} - f_\theta(x_n) \| \tag{1}$$

where $f_\theta(\cdot)$ denotes the embedding mapping, and $\theta$ is the trainable weights of the given neural network. $C$ and $y_n$ separately denote cluster centroid matrix with shape $d \times k$ and label assignment to $n_{th}$ image in the dataset, where $d$, $k$ and $N$ separately denote the embedding dimension, cluster number and dataset size. For simplicity in the following description, $y_n$ is presented as an one-hot vector, where the non-zero entry denotes its corresponding cluster assignment.

**Representation Learning.** After pseudo label generation, the representation learning process is exactly the same with supervised manner. To this end, a trainable linear classifier $W$ is stacked on the top of main network and optimized with $\theta$ together, which can be formulated as:

$$\min_{\theta,W} \frac{1}{N} \sum_{n=1}^{N} l(y_n, W f_\theta(x_n)) \tag{2}$$

where $l$ is the loss function.

Certainly, a correct label assignment is beneficial for representation learning, even approaching the supervised one. Likewise, a disentangled embedding representation will boost the clustering performance. These two steps are iteratively alternated and contribute positively to each other during optimization.

**Analysis.** Actually, clustering is to capture the global data relation, which requires to save the global latent embedding matrix $E \in \mathbb{R}^{d \times N}$ of the given dataset. Taking $k$-means as an example, it uses $E$ to iteratively compute the cluster centroids $C$. Here naturally comes a problem. It is difficult to scale to the extremely large datasets especially for those with millions or even billions of images since the memory of $E$ is linearly related to the dataset size. Thus, an existing question is, how can we group the images into several clusters without explicitly using global relation? Also, another slight problem is, the classifier $W$ has to reinitialize after each clustering and train from scratch, since the cluster IDs are changeable all the time, which makes the loss curve fluctuated all the time even at the end of training.

### 3.2 Unsupervised Image Classification

From the above section, we can find that the two steps in deep clustering (Eq.1 and Eq.2) actually illustrate two different manners for images grouping, namely clustering and classification. The former one groups images into clusters relying on the similarities among them, which is usually used in unsupervised learning. While the latter one learns a classification model and then directly classifies them into one of pre-defined classes without seeing other images, which is usually used in supervised learning. For the considerations discussed in the above section, we can't help to ask, why not directly use classification model to generate pseudo labels to avoid clustering? In this way, it can integrate these two steps pseudo label generation and representation learning into a more unified framework. Here pseudo label generation is formulated as:

$$\min_{y_n} \frac{1}{N} \sum_{n=1}^{N} l(y_n, f'_{\theta'}(x_n)) \ \ s.t. \ y_n \in \{0,1\}^k, y_n^T \mathbf{1}_k = 1 \tag{3}$$

where $f'_{\theta'}(\cdot)$ is the network composed by $f_\theta(\cdot)$ and $W$. Since cross-entropy with softmax output is the most commonly-used loss function for image classification,

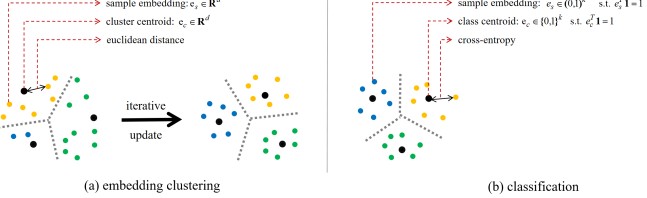

(a) embedding clustering                    (b) classification

**Fig. 2.** The difference and relation between embedding clustering and classification.

Eq.3 can be rewritten as:

$$y_n = p(f'_{\theta'}(x_n)) \qquad (4)$$

where $p(\cdot)$ is an arg max function indicating the non-zero entry for $y_n$. Iteratively alternating Eq.4 and Eq.2 for pseudo label generation and representation learning, can it really learn a disentangled representation? Apparently, it will easily fall in a local optima and learn less-representative features. The breaking point is data augmentation which is the core of many supervised and unsupervised learning algorithms. Normally, data augmentation is only adopted in representation learning process. However, this is not enough, which can not make this task challenging. Here data augmentation is also adopted in pseudo label generation. It brings disturbance for pseudo label, and make the task challenging enough to learn more robust features. Hence, Eq.4 and Eq.2 are rewritten as:

$$y_n = p(f'_{\theta'}(t_1(x_n))) \qquad (5)$$

$$\min_{\theta'} \frac{1}{N} \sum_{n=1}^{N} l(y_n, f'_{\theta'}(t_2(x_n))) \qquad (6)$$

where $t_1(\cdot)$ and $t_2(\cdot)$ denote two different random transformations. For efficiency, the forward pass of label generation can reuse the forward results of representation learning in the previous epoch. The entire pipeline of our proposed framework is illustrated in Fig.1. Since our proposed method is very similar to the supervised image classification in format. Correspondingly, we name our method as unsupervised image classification.

Compared with deep clustering, our method is more simple and elegant. It can be easily scaled to large datasets, since it does not need global latent embedding of the entire dataset for image grouping. Further, the classifier $W$ is optimized with the backbone network simultaneously instead of reinitializing after each clustering. To some extent, our method makes it a real end-to-end training framework.

### 3.3   Interpretation

**The Relation with Embedding Clustering.** Embedding clustering is the key component in deep clustering, which mainly focuses on three aspects: 1)

sample embedding generation, 2) distance metric, 3) grouping manner (or cluster centroid generation). Actually, from these aspects, using image classification to generate pseudo labels can be taken as a special variant of embedding clustering, as visualized in Fig.2. Compared with embedding clustering, the embedding in classification is the output of softmax layer and its dimension is exactly the class number. Usually, we call it the probability assigned to each class. As for distance metric, compared with the euclidean distance used in embedding clustering, cross-entropy can also be considered as an distance metric used in classification. The most significant point is the grouping manner. In $k$-means clustering, the cluster centroids are dynamicly determined and iteratively updated to reduce the intra-classes distance and enlarge the inter-classes distance. Conversely, the class centroids for classification are predefined and fixed as $k$ orthonormal one-hot vectors, which helps directly classify images via cross-entropy.

Briefly speaking, *the key difference between embedding clustering and classification is whether the class centroids are dynamicly determined or not.* In Deep-Cluster [2], 20-iterations $k$-means clustering is operated, while in DeeperCluster [3], 10-iterations $k$-means clustering is enough. It means that clustering actually is not that important. Our method actually can be taken as an 1-iteration variant with fixed class centroids. Considering the representations are still not well-learnt at the beginning of training, both clustering and classification cannot correctly partition the images into groups with the same semantic information. During training, we claim that it is redundant to tune both the embedding features and class centroids meanwhile. It is enough to fix the class centroids as orthonormal vectors and only tune the embedding features. Along with representation learning drived by learning data augmentation invariance, the images with the same semantic information will get closer to the same class centroid. What's more, compared with deep clustering, the class centroids in UIC are consistent in between pseudo label generation and representation learning.

**The Relation with Contrastive Learning.** Contrastive learning has become a popular method for unsupervised learning recently. Implicitly, unsupervised image classification can also be connected to contrastive learning to explain why it works. Although Eq.5 for pseudo label generation and Eq.6 for representation learning are operated by turns, we can merge Eq.5 into Eq.6 and get:

$$\min_{\theta'} \frac{1}{N} \sum_{n=1}^{N} l(p(f'_{\theta'}(t_1(x_n))), f'_{\theta'}(t_2(x_n))) \qquad (7)$$

which is optimized to maximize the mutual information between the representations from different transformations of the same image and learn data augmentation agnostic features. This is a basic formula used in many contrastive learning methods. More concretely, our method use a random view of the images to select their nearest class centroid, namely positive class, in a manner of taking the argmax of the softmax scores. During optimization, we push the representation of another random view of the images to get closer to their corresponding positive class. Implicitly, the remaining orthonormal $k$-1 classes will automatically

turn into negative classes. Since we use cross-entropy with softmax as the loss function, they will get farther to the negative classes during optimization. Intuitively, this may be a more proper way to generate negative samples. In normal contrastive learning methods, given an image I in a (large) minibatch , they treat the other images in the minibatch as the negative samples. But there exist the risk that the negative samples may share the same semantic information with I.

## 4    Experimental Results

### 4.1    Dataset Benchmarks and Network Architectures

We mainly apply our proposed unsupervised image classification to ImageNet dataset [31] without annotations, which is designed for 1000-categories image classification consisting of 1.28 millions images. As for network architectures, we select the most representative one in unsupervised representation learning, AlexNet [22], as our baseline model for performance analysis and comparison. It is composed by five convolutional layers for features extraction and three fully-connected layers for classification. Note that the Local Response Normalization layers are replaced by batch normalization layers. After unsupervised training, the performance is mainly evaluated by

- linear probes;
- transfer learning on downstream tasks.

Linear probes [40] had been a standard metric followed by lots of related works. It quantitatively evaluates the representation generated by different convolutional layers through separately freezing the convolutional layers (and Batch Normalization layers) from shallow layers to higher layers and training a linear classifier on top of them using annotated labels. For evaluation by linear probing, we conduct experiments on ImageNet datasets with annotated labels. Linear probes is a direct approach to evaluate the features learnt by unsupervised learning through fixing the feature extractors. Compared with this approach, transfer learning on downsteam tasks is closer to practical scenarios. Following the existing works, we transfer the unsupervised pretrained model on ImageNet to PASCAL VOC dataset [11] for multi-label image classification, object detection and semantic segmentation via fine-tuning. To avoid the performance gap brought by hyperparameter difference during fine-tuning, we further evaluate the representations by metric-based few-shot classification on $mini$ImageNet [34] without fine-tuning.

### 4.2    Unsupervised Image Classification

**Implementation Details.** Similar to DeepCluster, two important implementation details during unsupervised image classification have to be highlighted:

- Avoid empty classes;
- Class balance sampling.

**Table 1.** Ablation study on class number selection. Here we also report NMI t/labels, denoting the NMI between pseudo labels and annotated labels. FFT means further fine-tuning with fixed label assignments.

**Table 2.** Ablation study on whether data augmentation is adopted in pseudo label generation.

| Methods | Top1 Accuracy | | | NMI t/labels |
|---------|------|------|------|---------|
|         | conv3 | conv4 | conv5 | |
| UIC 3k | 41.2 | 41.0 | 38.1 | 38.5 |
| UIC 5k | 40.6 | 40.9 | 38.2 | 40.8 |
| UIC 10k | 40.6 | 40.8 | 37.9 | 42.6 |
| UIC 3k (FFT) | 41.6 | 41.5 | 39.0 | - |

| Methods | Aug | Top1 Accuracy | | |
|---------|-----|------|------|------|
|         |     | conv3 | conv4 | conv5 |
| UIC 3k | ✗ | 39.5 | 39.9 | 37.9 |
| UIC 3k | ✓ | 41.6 | 41.5 | 39.0 |

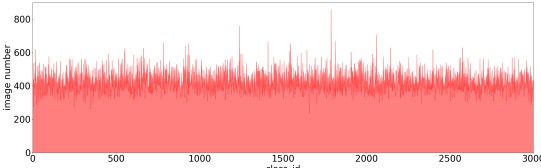

**Fig. 3.** Nearly uniform distribution of image number assigned to each class.

At the beginning of training, due to randomly initialization for network parameters, some classes are unavoidable to assign zero samples. To avoid trivial solution, we should avoid empty classes. When we catch one class with zero samples, we split the class with maximum samples into two equal partitions and assign one to the empty class. We observe that this situation of empty classes only happens at the beginning of training. As for class balance sampling, this technique is also used in supervised training to avoid the solution biasing to those classes with maximum samples.

**Optimization Settings.** We optimize AlexNet for 500 epochs through SGD optimizer with 256 batch size, 0.9 momentum, 1e-4 weight decay, 0.5 drop-out ratio and 0.1 learning rate decaying linearly. Analogous to DeepCluster, we apply Sobel filter to the input images to remove color information. During pseudo label generation and representation learning, we both adopt randomly resized cropping and horizontally flipping to augment input data. Compared with standard supervised training, the optimization settings are exactly the same except one extra hyperparameter, class number. Since over-clustering had been a consensus for clustering-based methods, here we only conduct ablation study about class number from 3k, 5k to 10k.

**Evaluation via Normalized Mutual Information.** Normalized mutual information (NMI) is the main metric to evaluate the classification results, which

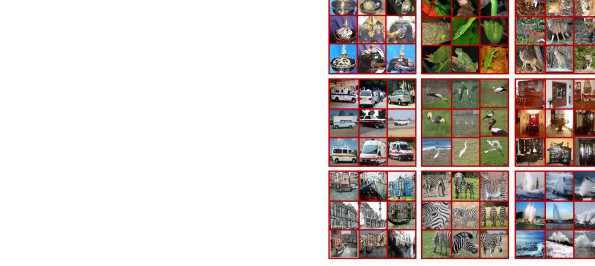

**Fig. 4.** Visualization of the classification results with low entropy.

ranges in the interval between 0 and 1. If NMI is approaching 1, it means two label assignments are strongly coherent. The annotated labels are unknown in practical scenarios, so we did not use them to tune the hyperparameters. But if the annotated labels are given, we can also use the NMI of label assignment against annotated one (NMI t/labels) to evaluate the classification results after training. As shown in the fifth column in Tab.1, when the class number is 10k, the NMI t/labels is comparable with DeepCluster (refer to Fig.2(a) in the paper [2]), which means the performance of our proposed unsupervised image classification is approaching to DeepCluster even without explicitly embedding clustering. However, the more class number will be easily to get higher NMI t/labels. So we cannot directly use it to compare the performance among different class number.

**Evaluation via Visualization.** At the end of training, we take a census for the image number assigned to each class. As shown in Fig.3, our classification model nearly divides the images in the dataset into equal partitions. This is a interesting finding. In the work of [1], this result is achieved via label optimization solved by *sinkhorn-Knopp algorithm*. However, our method can achieve the same result without label optimization. We infer that class balance sampling training manner can implicitly bias to uniform distribution. Furthermore, we also visualize the classification results in Fig.4. Our method can classify the images with similar semantic information into one class.

### 4.3 Linear Classification on Activations

**Optimization Settings.** We use linear probes for more quantitative evaluation. Following [40], we use max-pooling to separately reduce the activation dimensions to 9600, 9216, 9600, 9600 and 9216 (conv1-conv5). Freezing the feature extractors, we only train the inserted linear layers. We train the linear layers for 32 epochs with zero weight decay and 0.1 learning rate divided by ten at epochs 10, 20 and 30. The shorter size of the images in the dataset are resized to 256 pixels. And then we use 224×224 random crop as well as horizontal flipping to train the linear layer. After training, the accuracy is determined with 10-crops (center crop and four-corners crop as well as horizontal flipping).

**Table 3.** Linear probing evaluation on ImageNet. We mainly compare the performance of our method with DeepCluster. For reference, we also list the results of other methods.

| Methods | ImageNet | | | | |
|---|---|---|---|---|---|
| | conv1 | conv2 | conv3 | conv4 | conv5 |
| ImageNet labels | 19.3 | 36.3 | 44.2 | 48.3 | 50.5 |
| Random | 11.6 | 17.1 | 16.9 | 16.3 | 14.1 |
| DeepCluster [2] | 13.4 | 32.3 | 41.0 | 39.6 | 38.2 |
| SelfLabel $3k \times 1$ [1] | - | - | 43.0 | 44.7 | 40.9 |
| SelfLabel $3k \times 10$ [1] | 22.5 | 37.4 | 44.7 | 47.1 | 44.1 |
| **Ours** | **12.8** | **34.3** | **41.6** | **41.5** | **39.0** |
| Take a look at other self-supervised learning methods | | | | | |
| Contenxt [8] | 16.2 | 23.3 | 30.2 | 31.7 | 29.6 |
| BiGan [9] | 17.7 | 24.5 | 31.0 | 29.9 | 28.0 |
| Split-brain [40] | 17.7 | 29.3 | 35.4 | 35.2 | 32.8 |
| Jigsaw puzzle [28] | 18.2 | 28.8 | 34.0 | 33.9 | 27.1 |
| RotNet [13] | 18.8 | 31.7 | 38.7 | 38.2 | 36.5 |
| AND [20] | 15.6 | 27.0 | 35.9 | 39.7 | 37.9 |
| AET [38] | 19.3 | 35.4 | 44.0 | 43.6 | 42.4 |
| RotNet+retrieval [12] | 22.2 | 38.2 | 45.7 | 48.7 | 48.3 |

**Ablation Study on Class Number Selection.** We conduct ablation study on class number as shown in Tab.1. Different from DeepCluster, the performance 3k is slightly better than 5k and 10k, which is also confirmed by [1].

**Further Fine-Tuning.** During training, the label assignment is changed every epoch. We fix the label assignment at last epoch with center crop inference in pseudo label generation, and further fine-tune the network with 30 epochs. As shown in Tab.1, the performance can be further improved.

**Ablation Study on Data Augmentation.** Data augmentation plays an important role in clustering-based self-supervised learning since the pseudo labels are almost wrong at the beginning of training since the features are still not well-learnt and the representation learning is mainly driven by learning data augmentation invariance at the beginning of training. In this paper, we also use data augmentation in pseudo label generation. As shown in Tab.2, it can improve the performance. In this paper, we simply adopt randomly resized crop to augment data in pseudo label generation and representation learning.

**Comparison with Other State-of-The-Art Methods.** Since our method aims at simplifying DeepCluster by discarding clustering, we mainly compare our results with DeepCluster. As shown in Fig.3, our performance is comparable with DeepCluster, which validates that the clustering operation can be replaced

**Table 4.** Transfer the pretrained model to downstream tasks on PASCAL VOC dataset.

| Methods | Classification (%mAP) | | Detection (%mAP) | Segmentation (%mIU) |
|---|---|---|---|---|
| | FC6-8 | ALL | ALL | ALL |
| ImageNet Labels | 78.9 | 79.9 | 56.8 | 48.0 |
| Random-RGB | 33.2 | 57.0 | 44.5 | 30.1 |
| Random-Sobel | 29.0 | 61.9 | 47.9 | 32.0 |
| DeepCluster [2] | 72.0 | 73.7 | 55.4 | 45.1 |
| SelfLabeling $3k \times 10$ [1] | - | 75.3 | 55.9 | 43.7 |
| **Ours** | 76.2 | 75.9 | 54.9 | 45.9 |
| Take a look at other kinds of self-supervised methods | | | | |
| BiGan [9] | 52.5 | 60.3 | 46.9 | 35.2 |
| Contenxt [8] | 55.1 | 63.1 | 51.1 | - |
| Split-brain [40] | 63.0 | 67.1 | 46.7 | 36.0 |
| Jigsaw puzzle [28] | - | 67.6 | 53.2 | 37.6 |
| RotNet [13] | 70.87 | 72.97 | 54.4 | 39.1 |
| RotNet+retrieval [12] | - | 74.7 | 58.0 | 45.9 |

by more challenging data augmentation. Note that it is also validated by the NMI t/labels mentioned above. SelfLabel $[3k \times 1]$ simulates clustering via label optimization which classifies datas into equal partitions. However, as discussed above in Fig.3, our proposed framework also divides the dataset into nearly equal partitions without the complicated label optimization term. Therefore, theoretically, our framework can also achieve comparable results with SelfLabel $[3k \times 1]$, and we impute the performance gap to their extra augmentation. With strong augmentation, our can still surpass SelfLabel as shown in Tab.6. Compared with other self-supervised learning methods, our method can surpass most of them which only use a single type of supervisory signal. We believe our proposed framework can be taken as strong baseline model for self-supervised learning and make a further performance boost when combined with other supervisory signals, which will be validated in our future work.

## 4.4   Transfer to Downstream Tasks

**Evaluation via Fine-Tuning: Multi-label Image Classification, Object Detection, Semantic Segmentation on Pascal-VOC.** In practical scenarios, self-supervised learning is usually used to provide a good pretrained model to boost the representations for downstream tasks. Following other works, the representation learnt by our proposed method is also evaluated by fine-tuning the models on PASCAL VOC datasets. Specifically, we run the object detection task using fast-rcnn [14] framework and run the semantic segmentation task using FCN [26] framework. As shown in Tab.4, our performance is comparable with other clustering-based methods and surpass most of other SSL methods.

**Table 5.** Evaluation via few-shot classification on the test set of *mini*ImageNet. Note that 224 resolution is center-cropped from 256 which is upsampled from 84 low-resolutional images. It can be regarded as inserting a upsampling layer at the bottom of the network while the input is still 84×84. MP is short for max-pooling. For reference, the 5way-5shot accuracy of prototypical networks [32] via supervised manner is 68.2%.

| Methods | resolution | 5way-5shot accuracy | | | |
|---|---|---|---|---|---|
| | | conv3 | conv4 | conv5 | conv5+MP |
| UIC 3k | 224×224 | 48.79 | 53.03 | 62.46 | 65.05 |
| DeepCluster | 224×224 | 51.33 | 54.42 | 60.32 | 65.04 |
| UIC 3k | 84×84 | 52.43 | 54.76 | 54.40 | 52.85 |
| DeepCluster | 84×84 | 53.46 | 54.87 | 49.81 | 50.18 |

**Evaluation without Fine-Tuning: Metric-based Few-shot Image Classification on *mini*ImageNet.** Few-shot classification [34, 32] is naturally a protocol for representation evaluation, since it can directly use unsupervised pre-trained models for feature extraction and use metric-based methods for few-shot classification without any finetuning. It can avoid the performance gap brought by fine-tuning tricks. In this paper, we use Prototypical Networks [32] for representation evaluation on the test set of *mini*ImageNet. As shown in Tab.5, our method is comparable with DeepCluster overall. Specifically, our performances in highest layers are better than DeepCluster.

## 5    More Experiments

In the above sections, we try to keep training settings the same with DeepCluster for fair comparison. Although achieving SOTA results is not the main starting point of this work, we would not mind to further improve our results through combining the training tricks proposed by other methods.

### 5.1    More Data Augmentations

As discussed above, data augmentation used in the process of pseudo label generation and network training plays a very important role for representation learning. Recently, SimCLR[4] consumes lots of computational resources to do a thorough ablation study about data augmentation. They used a strong color jittering and random Gaussian blur to boost their performance. We find such strong augmentation can also benefit our method as shown in Tab.6. Our result in conv5 with a strong augmentation surpasses DeepCluster and SelfLabel by a large margin and is comparable with SelfLabel with 10 heads. Note that the results in this section do not use further fine-tuning.

### 5.2    More Network architectures

To further convince the readers, we supplement the experiments of ResNet50 (500epochs) with the strong data augmentation and an extra MLP-head pro-

**Table 6.** More experimental results with more data augmentations.

| Methods | Arch | ImageNet | | | | |
|---|---|---|---|---|---|---|
| | | conv3 | conv4 | conv5 | NMI | t/labels |
| DeepCluster [2] | AlexNet | 41.0 | 39.6 | 38.2 | - | |
| SelfLabel $3k \times 1$ [1] | AlexNet | 43.0 | 44.7 | 40.9 | - | |
| SelfLabel $3k \times 10$ [1] | AlexNet+10heads | 44.7 | 47.1 | 44.1 | - | |
| UIC (Ours) | AlexNet | 41.6 | 41.5 | 39.0 | 38.5 | |
| UIC + strong aug (Ours) | AlexNet | 43.5 | 45.6 | 44.3 | 40.0 | |

**Table 7.** More experimental results with more network architectures.

| Methods | Arch | Top-1 | NMI t/labels |
|---|---|---|---|
| Jigsaw [21] | Res50 | 38.4 | - |
| Rotation [21] | Res50 | 43.8 | - |
| InstDisc [35] | Res50 | 54.0 | - |
| BigBiGAN [10] | Res50 | 56.6 | - |
| Local Agg. [41] | Res50 | 60.2 | - |
| Moco [15] | Res50 | 60.6 | - |
| PIRL [27] | Res50 | 63.6 | - |
| CPCv2 [17] | Res50 | 63.8 | - |
| SimCLR [4] | Res50 + MLP-head | 69.3 | - |
| Mocov2 [6] | Res50 + MLP-head | 71.1 | - |
| SelfLabel $3k \times 10$ [1] | Res50+10heads | 61.5 | - |
| UIC + strong aug (Ours) | VGG16 | 57.7 | 46.9 |
| UIC + strong aug (Ours) | Res50 | 62.7 | 50.6 |
| UIC + strong aug (Ours) | Res50 + MLP-head | 64.4 | 53.3 |

posed by SimCLR[4] (we fix and do not discard MLP-head when linear probing).
As shown in Tab.7, our method surpasses SelfLabel and achieves SOTA results
when compared with non-contrastive-learning methods. Although our method
still has a performance gap with SimCLR and MoCov2 ($>>$500epochs), our
method is the simplest one among them. We believe it can bring more improve-
ment by appling more useful tricks.

# 6   Conclusions

We always believe that the greatest truths are the simplest. Our method validates
that the embedding clustering is not the main reason why DeepCluster works.
Our method makes training a SSL model as easy as training a supervised image
classification model, which can be adopted as a strong prototype to further
develop more advanced unsupervised learning approaches. We make SSL more
accessible to the community which is very friendly to the academic development.

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
