# OpenReview forum: "Unsupervised Image Classification for Deep Representation Learning"
_thecvf.com/ECCV/2020/Workshop/VIPriors — VIPriors Poster_

### Official Review · AnonReviewer2 · 2020-07-22
**Simple and effective method for unsupervised feature learning**

**Confidence:** 4
**Rating:** 8

**Review:**

#### 1. [Summary] In 2-3 sentences, describe the key ideas, experiments, and their significance.
The paper proposes a simple method for unsupervised deep clustering by iteratively (1) generating pseudo-labels by performing a forward pass through a CNN and (2) training the CNN using the generated pseudo-labels. The method is evaluated on ImageNet and performs competitively with other unsupervised learning methods.

#### 2. [Strengths] What are the strengths of the paper? Clearly explain why these aspects of the paper are valuable.
* The method is surprisingly simple and seems to perform competitively with clustering in latent space, which is much more computationally expensive.
* The paper is well written and easy to understand.
* The performed experiments are sound and demonstrate the effectiveness of the method.

#### 3. [Weaknesses] What are the weaknesses of the paper? Clearly explain why these aspects of the paper are weak.
* The bold numbers in Table 3 are rather misleading as they do not actually denote the best performance.

#### 4. [Overall rating] Paper rating
* 8: Top 50% of accepted papers, clear accept

#### 5. [Justification of rating] Please explain how the strengths and weaknesses aforementioned were weighed in for the rating.
The method is simple and effective and the paper is well written.

---

### Official Review · AnonReviewer1 · 2020-07-27
**Unsupervised Image Classification for Deep Representation Learning**

**Confidence:** 3
**Rating:** 6

**Review:**

1. [Summary] In 2-3 sentences, describe the key ideas, experiments, and their significance.

Authors in this paper described a modification of the embedding clustering method (DeepCluster) presented in [2]. Different from DeepCluster, this work proposes a unified pipeline, where clustering is directly performed using an image classification task. Experiments show first that their method achieves same or better performance than that of DeepCluster.

2. [Strengths] What are the strengths of the paper? Clearly explain why these aspects of the paper are valuable.

The authors found out that the embedding clustering phase in the previous method DeepCluster [2] can be avoided and its two phases can directly be performed using a classification task.
The method is pretty simple but it obtains state-of-the-art results.

3. [Weaknesses] What are the weaknesses of the paper? Clearly explain why these aspects of the paper are weak.

The paper seems a bit difficult to read. I found it a bit difficult to follow the story.
The whole paper is built on top of the work of [2], when considering the results, it could be described as a contribution to the field by itself.
Although authors propose to use different visual data augmentation (at the beginning only random crop and then further extending it with SimCLR techniques), and claim efficiency as they avoid storing embeddings, the work does not seem to be very much related to study visual inductive priors.

4. [Overall rating] Paper rating.

6.

5. [Justification of rating] Please explain how the strengths and weaknesses aforementioned were weighed in for the rating.

With a simpler method, the paper shows good results. However, readability, structure and out of scope possibility lower the score.

6. [Detailed comments] Additional comments regarding the paper (e.g. typos or other possible improvements you would like to see for the camera-ready version of the paper, if any.)

I suggest the authors to increase the size of Fig. 1, 2 and 4.

---

### Decision · Program_Chairs · 2020-07-29

**Decision:**

Accept (Poster)

**Comment:**

It is our pleasure to inform you that your paper has been accepted to the poster track of 1st Visual Inductive Priors for Data-Efficient Deep Learning Workshop.

Please note the following deadlines:
* August 11, 2020 - workshop material, including:
 * paper in PDF format;
 * pre-recorded video presentation;
 * slides of the presentation in PDF.
* September 15, 2020 - camera-ready paper

The reviews can be found on OpenReview. Please take these comments and suggestions into account when preparing the camera-ready version of your paper, which is due September 15, 2020. The camera-ready paper should be uploaded to OpenReview.

As part of the workshop, each accepted paper must submit a pre-recorded 90 second talk before August 11, 2020. You will receive more information on how to upload the material shortly. The requirements for the video are:
* Duration: maximum 90 seconds
* MP4 format
* File size max. 100 MB
* Has an inset with a video of the speaker
* 16:9 aspect ratio (strongly preferred)
* 1920x1080 resolution (strongly preferred, at least 720 height)

Our suggested software for pre-recording your presentation is Zoom. For more information, please refer to the following guides:
How to record with Zoom Guide: http://homepages.inf.ed.ac.uk/rbf/ECCV2020HowtoRecordusingZoom.pdf
How to Record with Zoom tutorial: https://www.youtube.com/watch?v=CR199W7HdC0
Please ensure that at least one of the authors of the paper is available to attend the workshop during the allotted times. Note that the workshop will take place in two sessions spread across time zones (details are to follow). We will send instructions on how to connect to the workshop as soon as possible. The schedule for all talks and papers will be posted soon at the workshop website: https://vipriors.github.io.

We look forward to seeing you at the workshop!